# Incorporating Context into BIM-Derived Data—Leveraging Graph Neural Networks for Building Element Classification

**Guy Austern** [1,*] **, Tanya Bloch** [2] **and Yael Abulafia** [1]

1    Faculty of Architecture and Town Planning, Technion—Israel Institute of Technology, Haifa 3200003, Israel; yael.abu@campus.technion.ac.il
2    Faculty of Civil and Environmental Engineering, Israel Institute of Technology, Haifa 3200003, Israel; bloch@technion.ac.il
*    Correspondence: guyaustern@technion.ac.il; Tel.: +972-545461918

**Abstract:** The application of machine learning (ML) for the automatic classification of building elements is a powerful technique for ensuring information integrity in building information models (BIMs). Previous work has demonstrated the favorable performance of such models on classification tasks using geometric information. This research explores the hypothesis that incorporating contextual information into the ML models can improve classification accuracy. To test this, we created a graph data structure where each building element is represented as a node assigned with basic geometric information. The connections between the graph nodes (edges) represent the immediate neighbors of that node, capturing the contextual information expressed in the BIM model. We devised a process for extracting graphs from BIM files and used it to construct a graph dataset of over 42,000 building elements and used the data to train several types of ML models. We compared the classification results of models that rely only on geometry, to graph neural networks (GNNs) that leverage contextual information. This work demonstrates that graph-based models for building element classification generally outperform classic ML models. Furthermore, dividing the graphs that represent complete buildings into smaller subgraphs further improves classification accuracy. These results underscore the potential of leveraging contextual information via graphs for advancing ML capabilities in the BIM environment.

**Keywords:** machine learning; graph neural networks; building information modeling; classification; semantic enrichment; graph classification

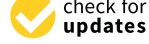



## 1. Introduction

The possibility to integrate machine learning (ML) in architecture, engineering, and construction (AEC) introduced numerous opportunities for advancement in design, building, and operation. The ability to process vast amounts of AEC data can support intelligent decision making in this complex field. Data, specifically machine-readable data, are the key to enabling ML applications.

The introduction of building information models (BIMs) has established a standardized framework for data representation in a machine-readable form, facilitating efficient modifications, queries, sharing, and analysis. However, effective utilization of these advantages is highly dependent upon the accurate representation of information within the BIM file. In practice, the accuracy of the information in a BIM file is not guaranteed. Thus, a prerequisite for using many advanced applications is the processing of the file to assure the quality of information and its adequate representation for the intended use. Semantic enrichment (SE) of BIM models has been suggested [1] in response to these challenges. SE is a goal-driven technique that aims to infer implicit information and provide an explicit representation of that information in accordance with the expected information structure of a receiving platform. Object classification plays a crucial role in this context, as accurately

identifying building elements serves as the foundation for various applications. Correct object classification is imperative where the original BIM file contains errors, corrupt, or incomplete classification information. The demand for automated enrichment is underscored by the absence of a standardized representation of semantic information in widely used modeling software such as Autodesk Revit [2].

Although the provided semantic information is often inaccurate, the geometric and topologic representation of the building elements are usually accurately provided. In recent academic studies, various methodologies have been suggested for enriching building information by utilizing the geometric characteristics of building elements as input [3]. These approaches span a broad spectrum of techniques, including heuristics and machine learning. These are comprehensively discussed in the background section of this work. While these methodologies have demonstrated noteworthy accomplishments, it is crucial to recognize that the topological aspect can also be harnessed for element classification, potentially resulting in enhanced accuracy.

We present a novel approach to enhance this accuracy by incorporating contextual information using geometric proximity graphs (GPGs) to represent building elements. GPG representations, such as Voronoi and Delaunay, provide an elegant solution to various computational challenges. In this context, we propose that a building model be translated into a GPG in which neighboring building elements, defined as intersections between elements, are connected by a graph edge. The nodes in the GPG of the building also contain a description of the element's basic features used in other classification methods.

The resulting GPG can be used as the basis for a graph neural network (GNN). GNN is a relatively recent development, allowing ML applications directly to graph-based data structures instead of Euclidean 2D information [4]. GNNs have demonstrated efficacy in enhancing predictions across various domains, including social networks, molecules, proteins, and knowledge graphs. Their performance excels in scenarios where the relationships between elements carry significant predictive implications.

The primary objective of this research was to investigate the potential enhancement of building element classification through the incorporation of contextual information. Our hypothesis posited that proximity relationships among building elements could function as indicators of the elements' identity or class, thus presenting valuable insights into improving the predictive capabilities of ML models.

To test this hypothesis, we extracted a dataset of over 35,000 building elements sourced from 11 different BIM files. Our novel extraction method enabled us to store proximity relations as well as geometric features and construct a GPG from the BIM model. The dataset was used to train different ML models to classify the basic categories of the BIM elements. We trained several types of classic ML models such as logistic regression and random forest as well as GNN models with different graph convolutional layers.

A comparison of the performance of the different models demonstrates that including contextual information through GNN models enhances predictive accuracy. Our results also reveal that among the tested GNN architectures, graph transformer networks yield the most favorable outcomes. Additionally, we show that partitioning the graph into subgraphs and employing graph classification techniques on these subsets substantially improves performance compared to the node classification task. Moreover, this subdivision into smaller data chunks exhibits practical advantages from an operational perspective.

Based on these results, we suggest that the incorporation of graph relations into conventional semantic enrichment processes can be seamlessly integrated into modeling workflows. These processes hold the potential for validating existing BIM information and reducing modeling errors. Furthermore, they can enable automatic translation between simplified 3D models and BIM software such as Autodesk Revit [2]. Finally, we envision a future BIM "autocorrect" functionality, which validates an element's type and properties using its geometry and topology.

## 2. Background

### 2.1. BIM and Semantic Enrichment

The emergence of BIM has ushered in a transformative paradigm in building design, where architectural information is stored as parametric objects with physical and material properties [5]. BIM technology has revolutionized the process of architectural design and construction [6], yielding profound benefits across diverse domains, including project management, economics, planning, interoperability, operational efficiency, and environmental considerations [7]. However, the abundance of semantic information generated in BIM models demands paying attention to data accuracy and consistency. It also introduces the interoperability problem as interpretations of semantic labels vary among stakeholders and software tools lead to inconsistencies, hindering seamless data exchange.

Contemporary BIM authoring tools originated in software developed at the end of the previous century when object-oriented design was the standard software paradigm [8]. The industry foundation class (IFC) standard is fundamental for this object-based data exchange. Although the IFC was designed to allow seamless collaboration and data exchange, in practice, when it comes to mapping entities and their relationships, the lack of universally enforced semantic classification and categorization protocols may lead to multiple data structures representing identical information. Different stakeholders may assign distinct semantic labels to similar objects, resulting in inconsistencies that pose a considerable challenge to interoperability [9]. For example, wide columns can be easily modeled using a wall element category. Diverse mapping protocols and classification methods employed by various BIM authoring tools further enhance the problem.

This variability in modeling semantics can lead to diminished data quality for further analysis. Moreover, the diverse classification methods employed by various authoring software further exacerbate issues related to interoperability and information exchange. The concept of semantic enrichment, as proposed by Belsky et al. [10], refers to generic expert systems that infer the semantic content of a model element. Enrichment processes were designed to address missing or erroneous information provided in the models, and to enhance interoperability between diverse software systems. Belsky proposed semantic enrichment for simplifying the process of semantic interpretation for receiving BIM applications, and reducing the need for domain-specific model view definitions. The approach was tested in a prototype platform called SeeBIM using precast concrete building models for automatically identifying precast joints, slab aggregations, etc. Later, the same SeeBIM platform was enhanced for enriching models obtained from point cloud data to support bridge inspection [11]. Bloch et al. [12] demonstrated the use of the SeeBIM engine for enriching BIM models in preparation for automated code checking (ACC), using a local regulatory code clause for security rooms in residential buildings.

Semantic enrichment has also been explored in the context of semantic web applications. As stated in [13], one of the reasons for using semantic web technologies is to enable logical inference through the use of first order logic (FOL) to infer new semantic information from the original building model. The major difference between this approach and the IFC-based enrichment, is in the representation of the building information, which in this case is a structured graph (resource description framework (RDF)).

As accurate identification of building elements is key to many possible applications (e.g., ACC, quantity takeoff, energy analysis, etc.), several researchers focused on automated BIM integrity checks through BIM object classification. For example, Koo et al. [14] used support vector machine to identify misclassified elements in the IFC files, but no justification for the choice of the algorithm was provided. Based on the work of [15], classification was identified as one of four types of semantic enrichment tasks (classification, association, calculation, and creation). As for the computational methods for enriching the models, Bloch and Sacks [15] suggest a dependency on the nature of the elements to be classified, but specific classification algorithms for each type of enrichment task were not suggested. This was later enhanced and fitted to semantic enrichment using a graph structure for representing building information [16], where the element classification task was tackled

using different ML algorithms, pointing to the superior performance of decision trees and random forests. Although the general direction of this work was to generate enriched BIM graphs to enable better collaboration and enhance interoperability, it seems an opportunity to leverage the graph structure for element classification was missed.

Other research endeavors focused on the classification of building elements, employing the geometric characteristics of elements extracted from BIM or IFC files as input. The system proposed by Belsky et al. [10] can infer the class of both objects and spaces based on their corresponding IFC properties. In the study conducted by Ma et al. [2], pairwise relationship matrices between objects are utilized to identify the objects. Their proposed method encapsulates the domain expert knowledge of the element classes, including relationships between elements, in the form of a matrix. Once the shape features and the spatial relationships of the objects to be classified are computed, the classification is performed using similarity calculation between the features of the models and those in the knowledge base. This work points to the importance of leveraging the relational aspect between the building elements.

Realizing that space functionality is key for advance procedures such as ACC, Bloch and Sacks [17] compared heuristic and feature-based ML methods in the context of space function classification and demonstrated that ML methods offer certain advantages in the context of semantic enrichment. However, their proposed approach is iterative, relying completely on geometric features in the first stage, and enhancing the results by adding connection features (access relationships between spaces), in the consecutive phases. This again points to the benefit of using relational information for classifying BIM elements. The same problem was later explored as a graph-based classification task where graph neural networks (GNNs) proved to be useful [18] and reached almost 80% classification accuracy with no need for iterations.

Kim et al. [19] trained convolutional neural networks (CNNs) to classify 2D images extracted from BIM objects. Koo et al. [20] utilized CNNs targeted for 3D object recognition, using either point clouds or multiple 2D views. They achieved improved classification of element subcategories such as door types surpassing a feature-based model—support vector machine (SVM). They emphasize the need for automatic extraction and processing of 3D objects from the BIM file to apply these methods. Finally, Wu et al. [21] compared the effectiveness of several classic feature-based machine learning models in predicting building element classes. They found that random forest (RF) models achieve the best classification results. However, their dataset is relatively small and does not originate from real BIM files. Additionally, they trained their models using BIM-derived features such as radial dimensions, which are not consistently available across all 3D authoring software platforms.

Despite initial successes in the field of ML for semantic enrichment, several research opportunities persist: model predictions can be improved, particularly when working with data derived from real BIM files. There is a need to broaden the range of types identified to accommodate the variety of architectural elements. Furthermore, the data acquisition process for enrichment can be generalized (in the sense that it should rely only on basic geometric features) and streamlined (in the sense that the data extraction from the original file should be efficient).

Many semantic enrichment efforts, aimed at enhancing interoperability and supporting various design analysis platforms, are based on parsing the IFC file. Although the potential benefits of graph representations for semantic enrichment have previously been suggested, to the best of the authors' knowledge, a graph-based learning approach for BIM element classification has not been demonstrated before. Furthermore, the added value of using contextual knowledge has not been compared to geometry-based classification approaches.

### 2.2. Buildings as Graphs

Graphs are mathematical constructs comprised of nodes and edges that represent relationships between these nodes [22]. They are effective tools for describing sets of entities,

ranging from molecules to social networks [23]. In their seminal work "the social logic of space", Hillier and Hansen [24] introduced the concept of utilizing graphs to represent architectural topological information, specifically the spatial relationships between rooms and their connectivity. Martin et al. [25] conducted a comprehensive review of various approaches to represent such topological information. They emphasize its importance in informing design decisions. Langenhan et al. [26] proposed the adoption of topological graphs to facilitate the retrieval of building information. Strug [27] applied topological graphs to evaluate design quality. Isaac et al. [28] described how BIM and graph theory could be integrated, as a preliminary step towards defining a model which represents building topology. Topological graph representations offer easily describe movement within a building, and consequently, have been employed for the analysis of building security [29], access control [30], and accessibility [31].

A different topological approach focuses on the physical components of the building and employs graph data structures to represent their relationships. By leveraging the inherent neighbor-finding capabilities of graph data structures, these queries offer markedly improved efficiency compared to those relying on tabular data. Geometry, properties, and relationships between the elements, as defined in IFC, often form the foundation for extracting these graphs. These derived graphs contain a wealth of information and enable complex queries within the model. Various techniques for extracting graphs from IFC have been reported in the literature [32–34]. These graphs are used for rule checking [35], for data merging [36], for generative design of modular buildings [37], and for model auditing and quantity takeoffs [38].

GPGs are a special instance of these topological graphs. They use the graph structure to represent the geometric distance between elements. These elegant representations are often used in classic graph algorithms such as Delaunay, Voronoi, and many others. For this research, we opted for an extreme interpretation of proximity, where an edge means an actual intersection between two building elements. We chose this relation since all building elements must touch at least one other element, and because this relation is relatively easy to check using a geometric intersection algorithm.

Our hypothesis is that the intersection relations between building elements could be a valuable indicator for the class of the building element. A diagram depicting these relations and their distinct characteristics across various types of building elements is shown in Figure 1. We set out to see if we can improve the performance of element classification using these relations. It should be noted that at this point, our focus is exclusively on physical building elements such as walls, doors, floors, etc. Abstract entities such as spaces or zones are not in the scope of this work.

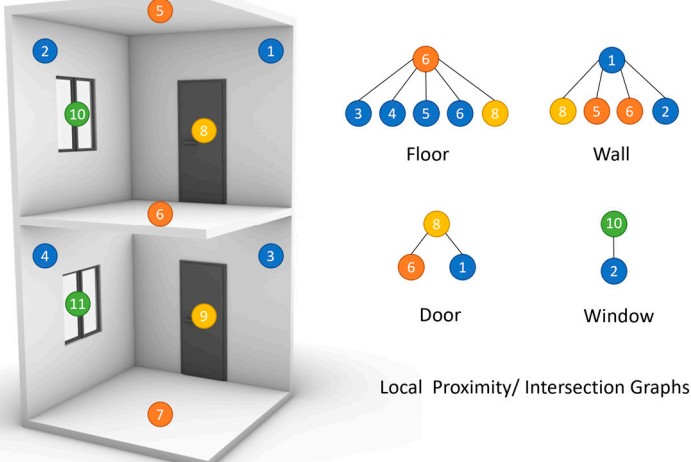

**Figure 1.** Local proximity graph of different types of building elements, colors signify the element types and number refer to specific elements in the design.

### 2.3. Classes of ML Models

ML is a branch of artificial intelligence (AI) dedicated to developing models that allow computers to learn from data and make predictions autonomously, without the need for explicit programming [39]. Within this highly popular domain, our research concentrates on three main types of ML models.

Classic machine learning, also known as traditional machine learning, comprises a collection of models and algorithms designed to predict an output label based on a set of input features. Classic machine learning techniques have been in extensive use since the previous century and have achieved success in various domains [40]. Among the most commonly used models in this category are the following: logistic regression (LR), originally proposed as a model for predicting drug potency [41], for which the coefficients of a logarithmic probability function are iteratively adjusted until the best prediction is reached [42]; support vector machines (SVMs), originally developed in Russia during the 1960s [43], are pattern recognition models that position a hyperplane between two groups in a dataset to maximize the margin between them; decision trees are models consisting of a sequence of binary tests leading to the correct classification branch [44]; and random forests (RFs), which employ an ensemble of decision trees with a voting mechanism [45]. These models exhibit exceptional performance in tasks such as classification, regression, clustering, and anomaly detection.

Artificial neural networks (ANNs) represent another subfield of ML that focuses on networks inspired by a simplified model of the human brain. These models can automatically learn to represent data through a training process involving forward and backward propagation of information [46]. Multi-layered perceptrons (MLPs) are one of the first examples of such networks. These feed-forward networks have been successfully utilized since the 1980s for tasks such as prediction, function estimation, and pattern recognition [47].

Deep learning (DL) networks are ANNs comprising a network architecture with several consecutive layers of neurons. These models gained popularity when [48] their successful application for character recognition was demonstrated. Later, AlexNet, a DL model with a convolutional neural network (CNN) architecture, achieved unprecedented results on the ImageNet image recognition dataset [49,50]. Due to the nature of the convolution operator, CNN-type models work best with data structured as 2D matrices, such as images. Other DL models excel in tasks such as natural language processing, speech recognition, and generative modeling.

Graph neural networks (GNNs) are a specialized type of neural network that is designed to work with graph-structured data. Bronstein et al. [4] describe the evolution of these networks to perform 2D convolutions on non-Euclidean data such as graphs, where traditional convolution operations are not well defined. In GNNs, information from neighboring graph nodes is embedded in the nodes when training the network, making them sensitive to the node's context. GNNs are highly effective in tasks that involve graph analysis, such as protein analysis, social networks, and knowledge graphs [51,52]. They can be trained to classify single nodes and edges of a graph, as well as entire graphs.

The use of GNNs in AEC is a relatively recent field of research. Recent studies include the use of GNNs to classify the function (as in use) of architectural spaces. This approach relies on a topological connectivity graph, in which rooms are connected to each other according to the existence of a physical passageway. Buruzs et al. [53] utilized graph convolutional networks (GCNs) on topology graphs extracted from IFC files to identify room functions in apartments. Wang et al. [18] used an improved GNN model called SAGE-E to identify room types in apartment layouts, achieving 79% accuracy. Similar graph representations were also used for an initial proof of concept for applying GNNs as the checking mechanism for design review, using accessibility requirements as a test case [54]. Yang and Huang [55] employed GNNs to classify the type of shopping mall, according to its entire connectivity graphs. However, these models focus primarily on classifying spaces, rather than building elements, which are the main concern of this work.

Focusing on building elements, Ouyang et al. [56] extracted a knowledge graph of the entire building from IFC representations of BIM models. They used this graph to associate

identical elements in different models of the same building (for example, structural and architectural) by matching their physical location as well as their relations on the graph. Collins et al. [57] based their work on the shape of 3D objects derived from an IFC file, and described it as a graph of physical vertices. These graphs were used as the input for GNNs which classify the 3D objects. They base their work solely on the object geometry without looking at its surroundings and reach an accuracy of 85%. Similarly, [58] used GNNs on graphs describing the geometry of two-dimensional planning zones, with a positive impact on the zone classification results. The potential of using contextual/topological information such as proximity graphs, together with geometric features, has yet to be explored, according to the best knowledge of the authors.

## 3. Research Aims

The main aim of this work is to demonstrate the ability of graph data to support possible ML applications in the AEC domain, and to compare the performance of classic and graph-based models. The scope of this paper is limited to the proof of concept for the idea that graph data which represent building models can support downstream ML applications. Within that, we focus our efforts on tasks from the semantic enrichment domain. More specifically, we aim to automatically classify building elements represented in BIM models to support the verification of information quality in BIM files. The underlying hypothesis behind this research is that adding contextual information to the geometric features can improve the results of classifying building elements.

## 4. Methods

The design of this research is described in Figure 2. Initially, we assembled an extensive dataset comprising over 42,000 building elements extracted from eleven BIM files. These files were obtained from local architect offices and encompassed both residential and commercial structures, reflecting real-world scenarios. To facilitate this endeavor, we devised a parametric methodology to extract the elements from the BIM files. Subsequently, the extracted elements were stored as JSON files, containing basic geometric details (e.g., bounding box dimensions, orientation, and number of parts). As part of the extraction process, we also identified and recorded the intersections between the elements, which were subsequently stored in the corresponding JSON files.

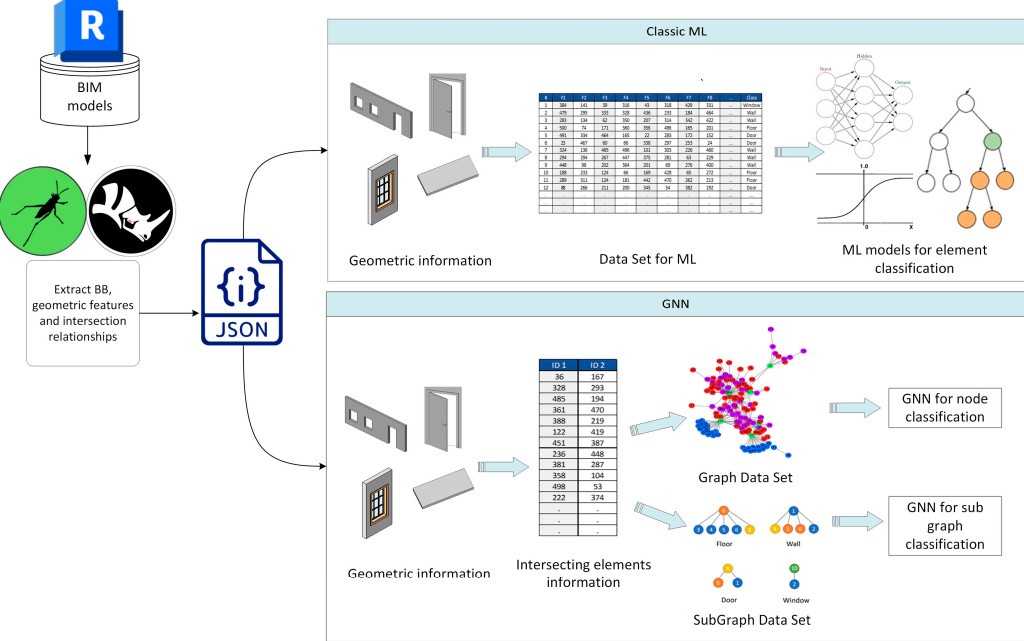

**Figure 2.** Desctption of the research method, starting with data extraction methods on the left and flowing to the ML methods on the right.

To establish a standardized classification process, the properties of the building elements were extracted from the JSON files and encoded as one-hot features within pandas data frames. This encoding scheme was employed to train multiple traditional machine learning (ML) models, aiming to classify the elements across the ten defined categories. Subsequently, GPGs were generated, incorporating the intersection relationships between the elements. Several GNN models were then trained using these graphs, employing various types of GNN node classification networks with different layer configurations. Furthermore, subgraphs capturing individual elements and their immediate neighbors were extracted from the larger graph, encoded, and stored separately. Various graph classification models were applied to these subgraphs, and the outcomes were compared to the previous node classification approach.

### 4.1. Graph DataSet Generation

The base requirement for conducting this research is a set of BIM models containing high-quality data about the element's classes, their geometric representation, and their relations to each other. In earlier studies [20,21], researchers generated such datasets by processing BIM files and extracting individual elements and their corresponding features. However, most datasets remain unpublished, and the ones that are accessible often lack the necessary proximity information crucial for our research. Hence, we constructed the dataset by extracting element information, and contextual information, from architectural models shared by local architects. A total of 10 Autodesk Revit files were obtained and translated to JSON as explained below. The models describe large residential and commercial buildings, which were all designed or built in the last few years. To increase the variance in the elements and avoid over-fitting, we chose to include only one building from each architect in the dataset.

The data extracted from the BIM models was organized into a JSON dataset, encompassing both the geometric features derived from the bounding box representation of each element, and the intersections between the elements. To prepare these data for ML implementation, we encoded the geometric features into binary vectors, ensuring compatibility with a wide range of ML models. We then tested several ML classification models to establish a baseline reference for assessing prediction quality. Notably, the classic models relied solely on the individual geometric features of each element, as these models cannot facilitate the representation of relational information. We then transformed the element dataset into a proximity graph, using the relational data, to leverage graph neural networks (GNNs) for the same element classification task. Several GNN models for node classification and for graph classification were implemented and evaluated. The resultant prediction outcomes from all the models were systematically compared, enabling us to gauge the influence of incorporating contextual information within this specific domain. Through this workflow, we aim to gain an understanding of how relational data can contribute to improving classification outcomes.

### 4.1.1. BIM to JSON

The first stage of the workflow is translating the obtained BIM files into machine-readable format via JSON. The conversion from Revit 2023 was performed using the Rhino Inside Revit plugin [59]. This plugin utilizes the strong functionality and well-known interface of the Grasshopper associative programming [60] platform inside the commercial Autodesk Revit [3] software. It was adopted for this research because of its dependability, customizability, and ease of use. Using the workflow illustrated in Figure 3, Revit elements of predetermined types (walls, doors, stairs, etc.) were imported into the Grasshopper environment using the Category Filter and Query Element modules. Then the Element Geometry was extracted and tested to ensure that the elements describe a physical object with a measurable volume. Textual attributes of the valid elements were accessed using the Inspect Element module and recorded into tables. These include Category Name, Family Name, Type Name, Material, Structural Behavior, and ID. Then, the element's orientation

was extracted using the Facing output of the Element Location module. The orientation was used to construct an oriented bounding box, which compactly envelopes the element and describes its basic geometric properties. Properties such as X, Y, and Z dimensions, volume, and relative height (in the model) were derived from the bounding box. Additionally, properties such as Solid Volume and Number of Components were extracted directly from the element and recorded.

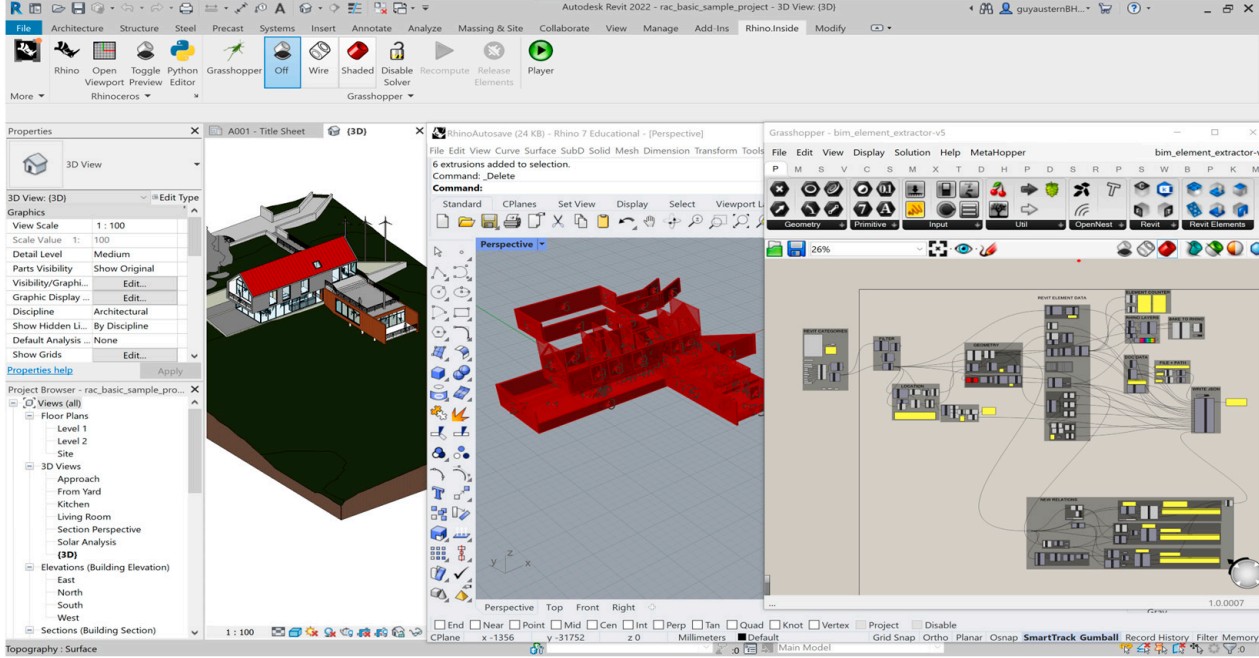

**Figure 3.** BIM element extraction process.

Finally, the recorded data were written on a JSON file, which is a commonly used dictionary format, independent of software language and operating system, using the following format:

```
[
    {
    "category_name": "Walls",
    "family_name": "Basic Wall",
    "type_name": "BO-EXT Con 65",
    "is_structural": "False",
    "BB_Y_dim": 389,
    "BB_Z_dim": 800,
    "BB_X_dim": 65,
    "relative_height": 0.24723022871239042,
    "BB_volume": 20240137
    "solid_volume": 13803420,
    "num_of_components": 1,
    "id": "700572",
    "unique_id": "6de6eb6a-bcbf-4e08-a866-d954b83f7159-000ab09c",
    "element_index": 6
    },
]
```

Each element was written separately on the file, with its properties recorded as key–value pairs. Additional information about the Revit file (*Name*, extraction date, geographic *Location*, and version number) was also appended to the file.

The JSON dataset consists of over 42,000 different elements of 10 classes: 'Walls', 'Doors', 'Furniture', 'Windows', 'Plumbing Fixtures', 'Floors', 'Stairs', 'Railings', 'Structural Columns', and 'Structural Framing'. The elements were derived from 10 different Revit models, consisting of residential apartment and office buildings. The division between the elements is shown in Table 1.

**Table 1.** Element categories in the dataset.

| Category | Count |
|---|---|
| Walls | 22,092 |
| Furniture | 5097 |
| Doors | 5072 |
| Windows | 3448 |
| Floors | 2132 |
| Plumbing Fixtures | 2710 |
| Structural Columns | 829 |
| Railings | 833 |
| Structural Framing | 421 |
| Stairs | 240 |
| **Total** | **42,874** |

### 4.1.2. Constructing the Relation Graph

We chose to represent each of the extracted BIM elements as nodes in a property graph. A property graph consists of a set of objects or nodes, and a set of edges connecting the objects. Both nodes and edges can also have multiple properties, which are represented as key–value pairs. Property graphs have been extensively used in domains such as IT operations, recommendation engines, and access control. Their structure has many benefits in terms of support for graph traversal algorithms and software development [61]. Property graphs are also widely used in commercial graph databases such as Neo4J, where they outperform traditional databases in several types of searches [62].

In our case, the property graph nodes are the BIM elements, the properties are their geometric features discussed in the previous section, and the edges represent proximity relations between elements.

We define the proximity relation between two elements as the existence of an intersection between their bounding boxes. In the common case of orthogonally oriented regular geometries, this means a tangible spatial intersection between the physical objects themselves. In the case of irregular objects, the notion of proximity may extend to situations where no actual physical contact between the elements exists. Still, this was the adopted approach due to the recognition of the substantial computational efficiency gains achieved by intersecting bounding boxes rather than the complex geometries of the actual elements.

In this stage of the extraction process, a geometric algorithm interrogated the relations between all the elements in the model. The Grasshopper *Clash* component was used to detect intersections between the bounding boxes of elements, as this component was designed to function in parallel on large datasets. The stability of the Grasshopper platform ensured that this process was successful, even though it involved checking all elements against all other elements. Even in huge REVIT building models with tens of thousands of elements, the extraction process was successful, although it processed for almost an hour on a 32GB RAM workstation with a dedicated graphic card. These relations, expressed as ID pairs, were also recorded, and stored. The relations were also written in the JSON file as lists of "unique_id" pairs.

We used the relation pairs to construct the property graphs in two different ways: (1) As a series of large graphs, one for each BIM file representing all the connected elements in the building, as shown in Figure 4. (2) As a series of smaller subgraphs, each centered on a single node which is connected to its first order neighbors, as shown in Figure 1.

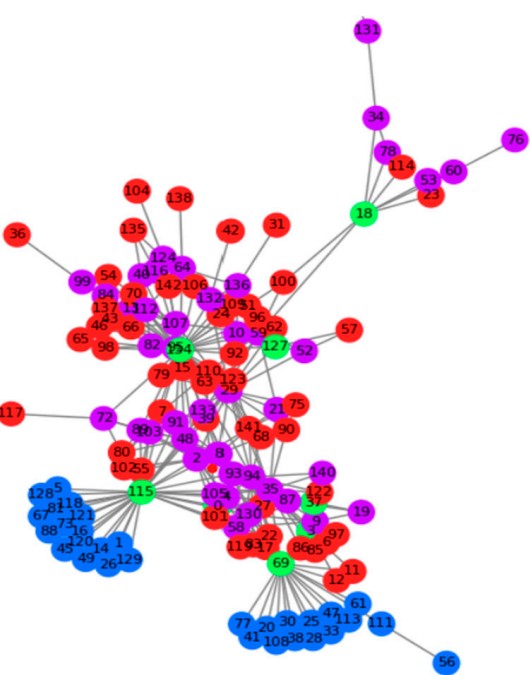

**Figure 4.** Graphic representation of a proximity graph derived from a small BIM file, color indictate element types and numbers refer to specific elemet ID's.

For representing the entire graph (1), we used the networkx [63] python library for graph creation and manipulation. Using this library, we first constructed a *Graph* object from the encoded elements and their connections. In this case, a few unconnected elements (with no intersections with other elements) were filtered out using the *connected_components* command.

For assembling the subgraphs (2), we also started with the connected networkx graph. In this version, we then traversed all the nodes in the Graph and used the *subgraph* command to split the graph into a first-order neighbor subgraph at each of the nodes. Each of the subgraphs was then recorded as a separate graph with all the information about the nodes. In addition, each subgraph entry contains a unique name originating from the unique ID (UID) of its root node as well as its category.

A possible future working scenario for this approach is one where the BIM elements are classified by an ML model immediately upon their creation. In this scenario, all surrounding elements will already have been classified, and the only question that remains is the identity of the new element. To model this scenario, we added the categories of the neighboring nodes to the encoding in the subgraph. To keep the encoding structure while masking the identity of the root node, its category was replaced with a generic "root" category. We saved this dataset separately under the name of "category encoding" and later ran the subgraph classification task both with and without the additional encoding. With this novel approach, the GNN was implemented for a subgraph classification task where all the contextual information, including the category of the neighbors, is already available to the model.

### 4.1.3. Feature Encoding

The initial JSON data were uploaded into a python runtime environment and stored as two lists, one of elements and their properties, and another with the intersection relations between the elements. The lists were then converted into pandas [64] data frames so that they could be used as input for ML models.

The element features were further processed to provide a dataset that can be utilized with all types of models. One-hot encoding (OHE) is a technique commonly used to represent input for deep learning models in many data science fields [65]. One-hot encoding

transforms labels into a binary vector in which every category is represented by a different index on the vector. This representation results in sparse vectors where most of the Booleans are false. Despite its large size, OHE has been known to improve the performance of many types of models, including regression models [66] and CNNs [67]. In GNN models, OHE is one of the most dominant encoding methods [68], where its uniformity is well adapted to the message-passing structure of the graph. To fit the data to our object of inquiry (GNNs), we chose to encode all features using OHE and used these data for training all the models mentioned in the next section, so that their performance can be compared.

Since the properties of the elements in our dataset are numeric, they had to be stored in labeled bins before encoding. This was achieved using the pandas *qcut* command, which divides data frame elements into a specified amount of roughly equal-sized bins. We divided all properties into 10 equally sized bins. This means that the bin boundaries are automatically set by the software according to the frequency of the appearance of dimensions in the dataset so that the bin sizes will be equal. Later investigation into the number of bins needed revealed that increasing the number of bins to 20 did not significantly change the results.

Table 2 shows the different bins generated for the X, Y, and Z dimensions of the element bounding box. The table shows how while the X and Y distributions are relatively uniform, the Z values are noticeably different. The bin division was saved on a separate file so that when analyzing a new model, the same bins could be used. Then, the pandas *get dummies* command was used to create frames in which all bins are represented as Boolean columns.

**Table 2.** Example bins resulting from the qcut command.

| Bounding Box X Dimension in cm | Bounding Box Y Dimension in cm | Bounding Box Z Dimension in cm |
| --- | --- | --- |
| 0.0 | 0.0 | 0.0 |
| 41.6 | 45.0 | 7.4 |
| 62.8 | 80.6 | 10.0 |
| 90.0 | 125.0 | 15.0 |
| 111.0 | 211.1 | 20.0 |
| 156.2 | 272.0 | 27.0 |
| 220.0 | 310.0 | 40.4 |
| 330.0 | 350.0 | 54.5 |
| 441.6 | 395.0 | 100.0 |
| 1001.7 | 1170.0 | 300.0 |

For example, when dividing all elements according to their bounding box X dimension, *qcut* identified that the lowest 10% of the elements fall between 0 and 41 cm, and created a label called "X_dim_0_to_41.6". Then the *get_dummies* command generated ten columns, one for each of the X dim bin labels, and assigned a TRUE or FALSE value to each of the elements according to its label. The process was then repeated for the following properties: bounding box X dimension, bounding box Y dimension, bounding box Z dimension, bounding box volume, solid volume, relative height, and the number of components in the BIM element. These properties were chosen for their simplicity and the fact that they can be easily measured regardless of the modeling platform. The seven properties resulted in a dataset with 70 features, used throughout the experiments.

### 4.2. Machine Learning with and without Contextual Information

The entire dataset was split into several parts for training and testing. Initially, one out of the ten BIM files was set aside as a test file, with the understanding that modeling conventions vary and that the true measure of any predictive model was on elements from a file it has never seen. This proved to be extremely influential on the model performance, which varied according to which of the files was set aside. For this reason, we performed all our experiments five times, each time setting a different model aside, training with the rest of the models and then testing the performance of the model on the sequestered file.

Data extracted from all files used for training was joined into a single dataset, element's order was randomized, and a standard 80/20 training/validation split was performed. When training on the graphs, the elements could not be mixed, and a random torch training/validation mask with the same split was applied instead.

### 4.2.1. Training Classic ML Models as a Baseline

To obtain an indication of the GNN performance in our scenario, we trained a series of classic ML models to perform element classification for comparison. We considered simpler models, which have been repeatedly and successfully used for a variety of purposes. For this purpose, we used the scikit-learn [69], a commonly used python library that contains many ML models and helper functions.

The encoded binary features of all the elements in the dataset were loaded into pandas data frames, with a Y field denoting their category labels but without the connection information. As discussed in the previous section, we used seven basic geometric features (size, volume, location, orientation, etc.) and encoded the features in them into ten bins and shuffled the data frame. We trained the models to predict the BIM category of the elements, using the scikit *fit* command on the training set, and used the *predict* command on the validation set (sourced from the same BIM files as the training set) to evaluate performance. Finally, we tested the generalization abilities of the models on the test set (sourced from previously unseen BIM files).

First, we trained a basic logistic regression model on the data as a baseline, and then tried other models, which were successfully used in the literature for this task: random forest, serial vector machine [21], and k-nearest neighbor classifiers. Seeking to compare the performance of the models, we trained them using the exact same framework and dataset, and used the most standard "out of the box" parameters possible for each of the models.

For ANN and GNN implementation, we adopted the pytorch [70] as the ML framework, due to its versatility, flexibility, and the availability of GNN architectures within this framework. For the pytorch version, we loaded the encoded element data from the JSON files into pytorch tensors, one for the features and another for the labels. The graph was converted into an adjacency matrix readable by the pytorch modules, comprised of pairs of connected elements referenced by their index in the torch data object.

The different tensors were loaded into a single pytorch Data object to be used by the models. We tested the pytorch environment by training a classic ANN—the multilayer perceptron (MLP) for the task. The MLP model was then built using the traditional three-layered architecture, with the input layer sized to fit the features, a hidden layer with the same size as the input, and an output layer sized to fit the classification categories. All layers were linear, fully connected layer. RELU activation + a dropout function was used between the first and the second layers, as well as between the second and the third. The results of the output layer were normalized using the log_softmax function.

The model was trained using the Adam optimizer with a standard learning rate of 0.01 and a decay of $5 \times 10^{-4}$. The loss function was defined as Cross Entropy loss. The model was trained for 200 epochs on the training data, and then tested both on the validation and the test dataset. These standard parameters were then fixed and maintained in all the training sessions to enable us to isolate the effect of the model architecture on the prediction accuracy.

### 4.2.2. Training a Graph Convolutional Network for Node Classification

We trained the GNN for the classification task using the same pytorch framework, data, and training parameters described above. The major difference between these models and the MLP mentioned above, was that the graph connectivity matrix was added to the dataset tensors. The connectivity matrix, representing the intersections between the building elements (edges in the graphs), enabled us to introduce contextual information into the node classification model. To utilize connectivity information, GNNs are built with specifically network layers that enable message passing between neighboring nodes.

We built the GNN model using the pytorch geometric (PyG) library [71]. PyG is an extension to the pytorch library that enables running deep learning models on irregular data structures such as graphs. The network was structured according to the classic design, as described in the original GCN paper [72]. This network with an input layer the same size as the features and a single hidden layer with an input of a similar size and an output size according to the classification classes. The input layer is followed by a RELU function, and the output results are normalized using a softmax function. The PyG library allowed us to replace the first two fully connected, feature sized layers from the MLP with several types of convolution layers designed specifically for graph structures. Several popular PyG graph convolution layers were explored for building the network for node classification tasks: GraphCONV, which implements the Weisfeiler–Leman graph isomorphism as described in [73]; SAGEConv [74], which leverages node feature to generate embeddings for previously unseen data; and TransformerConv [75] were all tested. All networks were trained on the same data as the previous models, and with the same training parameters as the MLP model.

### 4.2.3. Training a Subgraph Classification GNN

GNNs are often used to classify entire graphs, such as the ones represented by proteins or molecules [52]. Our basic hypothesis is that information about the immediate neighbors of an element is useful for identifying the element's class. The immediate neighborhood of each node can be represented by a simple graph containing the unclassified node neighbors. This compact representation is highly efficient when querying the model about a specific node. To test our hypothesis, we extracted the first-order neighborhood for each of the building elements and trained a GNN to classify these subgraphs. As the number of graphs dramatically increased, we divided the dataset into batches of 1000 for training purposes. The graph classification network was built using the same GNN layers described in Section 4.2.3, with a mean pooling layer to handle the batch training results.

## 5. Results

### 5.1. Comparison between Classification Results

We trained the different models described above using a typical 80–20 training/validation split. As previously mentioned, we trained the models on data derived from ten of the BIM files, and kept one file aside for testing, allowing us to see the performance of the model on previously unseen data. The results presented in Table 3 show the significant variance between the different ML models. It is evident that GNN-based models outperform classic ones. Amongst the GNN models, subgraph classification models with category encoding achieve the best performance, exceeding that of the classic models by over 5%. The table also shows how the prediction accuracy on the test file, a previously unseen BIM file, was substantially lower. This result is surprising, as previous studies [19,21] have shown that ML models are able to generalize BIM classification problems. We suggest that the decreased performance on the unseen file might be due to the source of the data used in our study. We used elements sourced from the BIM files of real buildings, which were modeled by different architecture offices, as opposed to the Revit demonstrator files used in [21]. This led to naturally occurring variations in modeling conventions, which are common in the AEC industry. We propose that the real performance of the models should be measured by their prediction accuracy on unseen files, as it reflects their behavior in the real world.

Further testing validated the observation regarding the varying prediction accuracy between different models, dependent upon the choice of the test files. We ran the whole train–validation–test setup five times, keeping a different file outside of the training data every time. Figure 5 shows the average performance of all the different classifiers on the five runs and illustrates the vast differences in prediction accuracy. The performance of all the models on the training and the validation set (which are hidden elements from the training files) is high and relatively stable, which supports findings from previous research. In contradiction, the performance on the elements from BIM models not included in the

training (the test files) is significantly lower and subject to a variance of over 5%. These differences raise questions about the ability of ML models to generalize BIM classification problems, given the small size of the existing datasets.

**Table 3.** Results of an example training process.

| Classic Models | Train | Validate | Test |
|---|---|---|---|
| KNNeighbors | 0.956 | 0.945 | 0.778 |
| Logistic Regression | 0.896 | 0.894 | 0.851 |
| Support Vector Machine | 0.966 | 0.961 | 0.835 |
| Random Forest | 0.977 | 0.965 | 0.793 |
| MLP | 0.948 | 0.942 | 0.815 |
| **GNN Node Classifiers** | | | |
| GraphConv | 0.954 | 0.944 | 0.853 |
| SAGEConv | 0.979 | 0.966 | 0.804 |
| TransformerConv | 0.986 | 0.973 | 0.845 |
| **GNN Subgraph Classification** | | | |
| GraphConv | 0.996 | 0.984 | 0.870 |
| SAGEConv | 0.996 | 0.986 | 0.876 |
| TransformerConv | 0.996 | 0.984 | 0.862 |
| **GNN Subgraphs with Category Encoding** | | | |
| GraphConv | 0.998 | 0.989 | 0.921 |
| SAGEConv | 0.998 | 0.989 | 0.903 |
| TransformerConv | 0.998 | 0.989 | 0.907 |

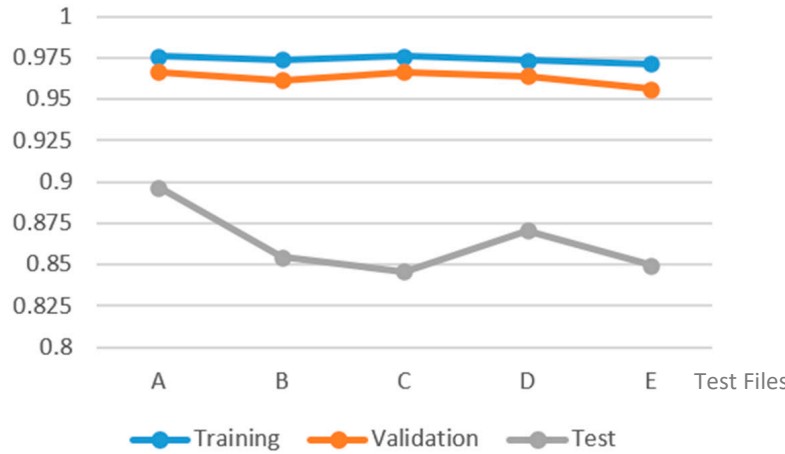

**Figure 5.** Prediction accuracy on different test files, averaged across all tested models.

To accurately assess the performance of the different models, we averaged their prediction accuracy over the five test files as shown in Figure 6. In terms of the performance of the different model families, the classic ML models have the poorest average performance. Within this family, the support vector machine model displayed the best with an accuracy of 86.7%. In the averaged results, the GNN node classifiers performed better than the classic models and the best-performing model in this family achieved 87.7%, demonstrating that contextual information can improve prediction accuracy. Here, we see significant variations between the different types of GNN layers used, and the type of classification task—node or subgraph classification. In general, the GNN subgraph classification models achieved 88.3%, a further improvement over the GNN node classification models.

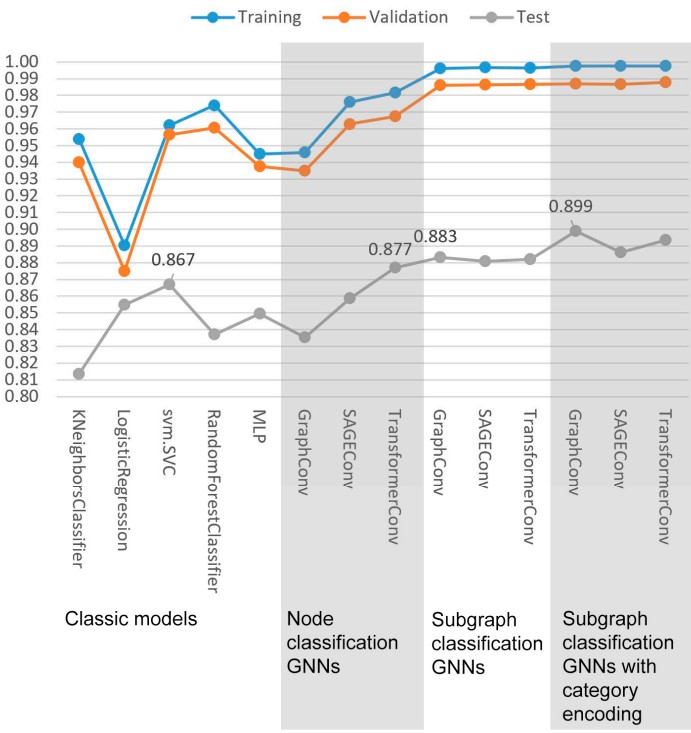

**Figure 6.** Average prediction accuracy of the different models.

Figure 6 also shows how adding the explicit representation of the categories of the neighboring nodes into the subgraphs dramatically improved the prediction results. With the additional information, GraphConv-based GNN networks achieved a prediction accuracy of 89.9%, more than 3% over the best-performing classic model.

### 5.2. Test Case Analysis

To improve our understanding of the results, one of the test cases was analyzed in detail. The test case is a 10 stories tall residential building with an irregular shape as shown in Figure 7. The building has four distinct floorplans, with a unique ground floor, penthouse, and levels 4–6. The building is constructed from a concrete skeleton with block infills and synthetic stone cladding. The BIM model includes all architectural and structural elements of the building, including basic furniture and plumbing fixtures, numbering 4442 elements in total.

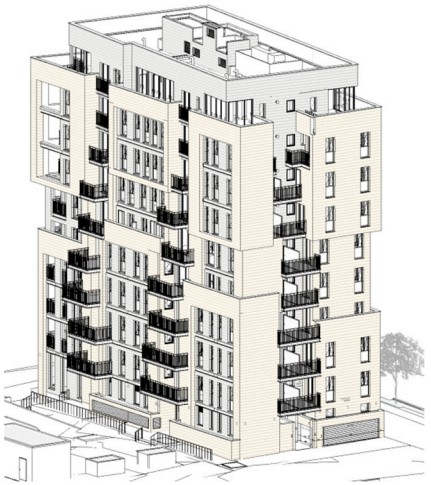

**Figure 7.** Residential case study. Image by O2A architecture.

The confusion matrixes describing the classification of the elements in the given model are presented in Figure 8. As seen in these matrixes, one of the biggest differences between the classic and graph-based models is the classification of windows as doors (common for the classic SVM model), or as plumbing fixtures (common for graph-based models).

| | | SVM | | | | | | | | | | GCN | | | | | | | | | | Sub Graphs | | | | | | | | | |
|---|---|---|---|---|---|---|---|---|---|---|---|---|---|---|---|---|---|---|---|---|---|---|---|---|---|---|---|---|---|---|---|---|
| | Walls | 1805 | 91 | 12 | 6 | 45 | 1 | 1 | 0 | 22 | 3 | 1894 | 42 | 7 | 15 | 7 | 9 | 0 | 2 | 2 | 0 | 1874 | 55 | 14 | 8 | 6 | 4 | 0 | 2 | 21 | 2 |
| | Doors | 0 | 536 | 0 | 0 | 74 | 0 | 0 | 0 | 0 | 0 | 10 | 447 | 0 | 0 | 101 | 0 | 0 | 2 | 0 | 0 | 3 | 529 | 7 | 0 | 21 | 0 | 0 | 0 | 0 | 0 |
| | Plumbing Fixtures | 0 | 0 | 131 | 0 | 1 | 175 | 0 | 0 | 0 | 0 | 0 | 0 | 138 | 2 | 9 | 71 | 0 | 15 | 0 | 0 | 0 | 0 | 208 | 0 | 10 | 5 | 0 | 12 | 1 | 0 |
| | Floors | 16 | 0 | 1 | 190 | 0 | 2 | 0 | 0 | 0 | 3 | 16 | 0 | 0 | 194 | 0 | 1 | 0 | 1 | 0 | 0 | 15 | 0 | 7 | 184 | 0 | 2 | 1 | 0 | 0 | 3 |
| Actual | Windows | 0 | 133 | 15 | 0 | 334 | 63 | 0 | 17 | 0 | 0 | 0 | 21 | 131 | 0 | 314 | 73 | 0 | 20 | 0 | 0 | 0 | 68 | 101 | 0 | 375 | 7 | 0 | 8 | 0 | 0 |
| | Furniture | 0 | 0 | 0 | 2 | 0 | 596 | 0 | 0 | 0 | 6 | 0 | 0 | 5 | 2 | 3 | 567 | 0 | 0 | 0 | 0 | 1 | 0 | 1 | 1 | 0 | 574 | 0 | 0 | 0 | 0 |
| | Stairs | 1 | 0 | 0 | 4 | 0 | 1 | 13 | 0 | 0 | 0 | 1 | 0 | 0 | 6 | 0 | 0 | 12 | 0 | 0 | 0 | 0 | 0 | 0 | 3 | 0 | 0 | 16 | 0 | 0 | 0 |
| | Railings | 0 | 25 | 0 | 1 | 1 | 1 | 0 | 99 | 0 | 0 | 0 | 7 | 0 | 1 | 4 | 1 | 1 | 112 | 0 | 0 | 0 | 7 | 1 | 1 | 4 | 0 | 1 | 112 | 0 | 0 |
| | StructuralColumns | 10 | 0 | 0 | 0 | 0 | 0 | 0 | 0 | 5 | 0 | 13 | 0 | 0 | 0 | 0 | 0 | 0 | 0 | 2 | 0 | 10 | 0 | 1 | 0 | 0 | 0 | 0 | 0 | 4 | 0 |
| | Structural Framing | 0 | 0 | 0 | 0 | 0 | 0 | 0 | 0 | 0 | 0 | 0 | 0 | 0 | 0 | 0 | 0 | 0 | 0 | 0 | 0 | 0 | 0 | 0 | 0 | 0 | 0 | 0 | 0 | 0 | 0 |
| | | Walls | Doors | Plumbing Fixtures | Floors | Windows | Furniture | Stairs | Railings | StructuralColumns | Structural Framing | Walls | Doors | Plumbing Fixtures | Floors | Windows | Furniture | Stairs | Railings | StructuralColumns | Structural Framing | Walls | Doors | Plumbing Fixtures | Floors | Windows | Furniture | Stairs | Railings | StructuralColumns | Structural Framing |
| | | | | | | | | | | | | | | | | Predicted | | | | | | | | | | | | | | | | |

**Figure 8.** Example of a confusion matrix for the different models. Darker green signifies higher element frequency.

To assess the models more closely and understand the reason for these differences, we inspected the individual elements of the BIM models where the prediction was false. In examining the classification outcomes, the occurrence of windows being misclassified as "doors" by the SVM model reveals a noteworthy difference in the behavior of the models. The windows with similar dimensions to doors, seemed to confuse the classic SVM model, leading to the mislabeling of 133 windows as doors. In contrast, the graph-based models, primarily guided via interconnections between building elements, demonstrate resilience against such geometric similarities. Figure 9 shows how the fact that windows usually neighbor only wall-type elements, while doors are also connected to floors helped the graph-based model differentiate between the elements. This highlights the strength of graph-based models in capturing contextual relationships.

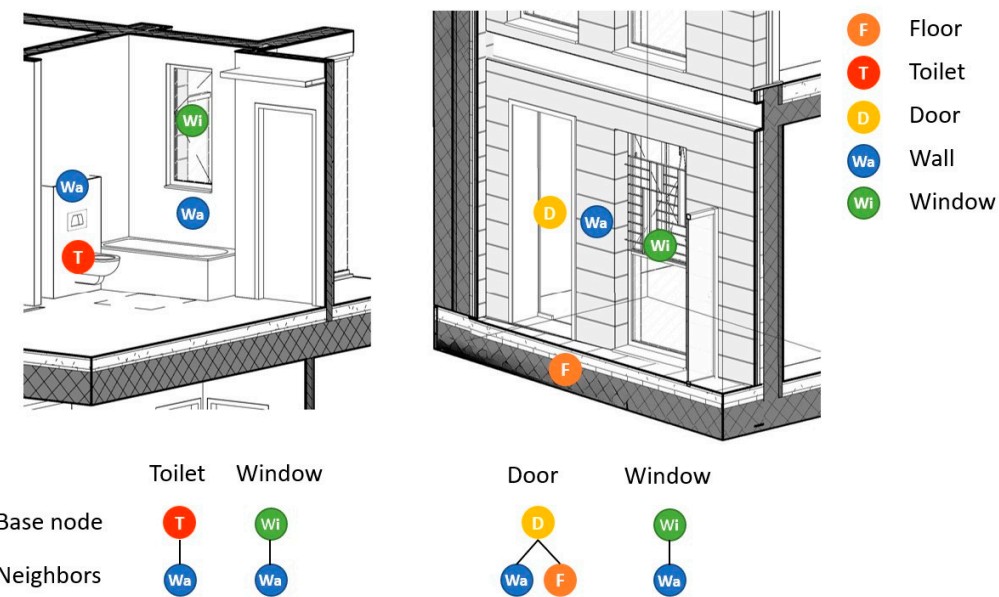

**Figure 9.** Left: toilet and window have different dimensions and similar context. Right: door and window have similar dimensions and different contexts.

On the other hand, graph-based models tended to classify windows as plumbing fixtures while the classic SVM model avoided this mistake. After a closer examination, we see the commonality—both windows and plumbing fixtures predominantly intersect with wall elements. This shared contextual information might contribute to the misclassification.

## 6. Discussion

In this work, we demonstrated the applicability of graph neural networks for classifying the elements in BIM models. In previous work, Wu et al. [21] demonstrated that random forest models can reach scores of over 99% on the BIM element classification task. Koo et al. [20] show how CNNs can identify element subtypes with and accuracy of 95%. Our findings demonstrated similar results on elements sourced from several BIM files. On these elements, our best-performing classification method consistently reached an accuracy of over 99% on the training data and 98% on the validation data. However, when classifying elements from BIM files that were not part of the training process, the accuracy of all the tested models dropped significantly to 81%–90%. Since previously unseen files (not elements) are the real targets of semantic enrichment processes, this raises a question about how BIM classification models are evaluated. We suggest that dividing elements into training and validation sets is not enough, and that they should be tested on elements from BIM files not included in the training process.

As presented in the results section, introducing context in the form of geometric proximity graphs improved the prediction accuracy in the element classification task on previously unseen BIM files. Breaking up the graph into smaller parts and classifying each one of them according to the properties and categories of their immediate neighbors achieved a further improvement in the results. Since this type of classification model does not require the entire relation graph of the building, this approach has an operational advantage due to its scalability.

### 6.1. Limitations

While the GNNs presented promising advancements, limitations in generalization and the need for diversity in the training dataset are acknowledged. The findings emphasize the significance of a diverse and high-quality dataset with a broad spectrum of complex geometries and contextual information to enhance model performance and accuracy. Our dataset contains 42,000 individual elements, from eleven different BIM files. All the training files come from the same country, with similar practices and building regulations. Local differences in construction methods and regulations might affect the dimensional features that traditional models are dependent on, for example, a concrete wall is dimensionally distinct from a wall made from wood or stone. To develop a general classification model, training information would need to include files from different countries, with different regulations, typologies, and construction methods.

However, the geometric relations between walls, floors, and doors are not dependent upon the construction technique used and remain invariant across geography, method, and even time. Our findings indicate that introducing contextual information in the form of intersection graphs between building elements assists the models in generalizing. By using these relations, GNN models improved the prediction accuracy on previously unseen BIM files sourced from different architectural offices. This suggests that graph-based GNNs can be more adept at generalization and may require smaller training sets than classic models.

### 6.2. Future Work

The graphs used in this research are based on "intersection" relations between different elements. Intersections are the simplest types of topological relations between elements and were found to contain information relevant to the classification task. However, these relations are not the only ones that can be utilized to improve classification. For example, a "co-verticality" relation, indicating that two elements share x and y coordinates with different z values, could be a strong indicator of structural properties. A "containment"

relation is likely the best indicator for elements such as windows, which are contained within walls, but less relevant for other elements. A more complex relation such as a "connected to" can provide significant information for classifying plumbing and electric building systems.

Future work can build upon these relations to improve and expand classification processes. It would be interesting to test which relations would be the best indicators of which properties of the elements. Furthermore, combining more than one type of relation into a single graph can potentially introduce even more information into the data. Current GNN models do not perform well on graphs with multiple edge types that represent different relations. However, GNNs are a rapidly advancing field and such an improvement may be already under development.

## 7. Conclusions

In this work, we presented a framework for processing BIM files into graph representation to leverage advanced ML algorithms applicable directly to graph data. We have demonstrated a workflow for extracting element features and intersection-based graphs of building elements from BIM models. The presented test case of BIM element classification is based on processing BIM files obtained from the industry. Over 42,000 elements, their features, and their relations were encoded and utilized as the basis for training several types of machine learning models to classify building elements.

A comparison between the performance of the different models on identical data shows that a GNN classification can improve the accuracy of the other tested models. The best-performing models used subgraph classification, where each element, its features, and the categories of its immediate neighbors were separated from the main graph and classified individually. These models achieved 89.9% accuracy in the classification task of elements from BIM files not used for training. This represents a significant improvement over classic ML models which reached a maximum accuracy of 86.7%.

We believe that the introduction of contextual information in the shape of graphs into BIM workflows can lead the way to advanced ML capabilities. Element classification and model enrichment illustrate how ML-powered BIM can accommodate automatic translation and data exchange between modeling software. In one possible application, users model a featureless box, and the authoring software automatically assigns its category and fills in design parameters (autocomplete). In another, the BIM software automatically checks every element added by the user for modeling errors (autocorrect). The local character and improved performance of subgraph classification models are well suited to the real-time integration of ML within BIM authoring software.

**Author Contributions:** Conceptualization, G.A. and T.B.; methodology, G.A. and T.B.; software, G.A. and Y.A.; formal analysis, G.A. and Y.A.; investigation, G.A., T.B. and Y.A.; resources, G.A, T.B. and Y.A.; data curation, Y.A.; writing—original draft preparation, G.A.; writing—review and editing, G.A. and T.B.; visualization, G.A., T.B. and Y.A.; supervision, G.A.; project administration, G.A.; funding acquisition, G.A. and T.B. All authors have read and agreed to the published version of the manuscript.

**Funding:** This research was funded by grant 880007 from the Israel Ministry of Construction and Housing.

**Data Availability Statement:** Restrictions apply to the availability of the data. Data were obtained from architectural offices and are available from the corresponding author with the permission of the architectural firms.

**Acknowledgments:** The authors would like to thank Yasha Jacob Grobman for his help in contacting the various architecture offices that contributed to the BIM models for the research. We would also like to thank the architects who agreed to contribute their data to this research. Finally, we would like to thank the research team that helped tag and prepare the data for learning: Abraham Shkolnik and Oz Kalo.

**Conflicts of Interest:** The authors declare no conflicts of interest. The funders had no role in the design of the study; in the collection, analyses, or interpretation of data; in the writing of the manuscript; or in the decision to publish the results.

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
