# Peer review of "Incorporating Context into BIM-Derived Data—Leveraging Graph Neural Networks for Building Element Classification"

_buildings, doi:10.3390/buildings14020527_

Round 1

Reviewer 1 Report

Comments and Suggestions for Authors

Dear Authors, 

your manuscript looks neglected, with many repeated acronyms definition. 

On the other side, there are several undefined terms. 

Please, correct them and make your manuscript pedant. Temporary, it looks pretty clumsy.

Comments on the Quality of English Language

Although I am not an English expert, I singed some phrases that sounds weird. in total, the manuscript is understandable.

Author Response

We thank the reviewer for this important comment. We revised the acronym definitions s requested. We also edited the grammar of the manuscript extensively as can be seen in the modified document.

Reviewer 2 Report

Comments and Suggestions for Authors

Keywords: Line 29: Please remove the placeholders or replace them with relevant keywords.

Introduction: Line 102: The introduction would benefit from a clear statement towards the end, summarizing the main aim and contribution of the research proposed.

Background:

Lines 241, 246, 248 etc.: Once an abbreviation is introduced and defined, it should be consistently used thereafter instead of the full term.

Lines 103-304: While the section cites various studies, it tends to list them rather than critically analyze their contributions or limitations. A more in-depth analysis of these studies, including how they directly relate to the research proposed, would enhance the quality of the manuscript. Moreover, addressing how the proposed research builds upon or differs from these cited works would provide a clearer understanding of the paper's contribution to the field.

Lines 105-113, 185-192: Repetition should be avoided. For instance, the explanation of BIM and its challenges is revisited multiple times. Consolidating these explanations would streamline the content.

Methods:

Lines 437-453 : While the process of feature encoding and model training is described, it might be helpful to briefly discuss why certain methods (e.g., One Hot Encoding) were chosen over others.

Lines 530-565: It would also be useful to clarify the reasoning behind the specific parameters and configurations used in the models (Logistic Regression, Random Forest, Serial Vector Machine, and K-Nearest Neighbor classifiers).

Figures: Lines 228, 317, 636, 645: Figure 2 is not referenced in the text. Additionally, the mention of a figure in the text should precede the figure itself. So please proceed to correct Figure 1. Last, there are two figures labeled as Figure 6 in the text.

Results: Lines 611-621: The discussion on the variance in prediction accuracy across different test files is insightful but could benefit from deeper analysis. Why did these variances occur, and what implications does this have for the generalizability of the models?

Discussion: Lines 664-715: While the section begins by summarizing the applicability of Graph Neural Networks, it would be beneficial to more directly connect these observations to the specific results presented earlier. For instance, discuss how the results support the advantages of using GNNs in BIM model classification. Also please elaborate on how the findings relate to existing literature.

Author Response

We thank the reviewer for these instructive comments. See answers to the specific comments in italic below

Keywords: Line 29: Please remove the placeholders or replace them with relevant keywords.

The Issue was addressed

Introduction: Line 102: The introduction would benefit from a clear statement towards the end, summarizing the main aim and contribution of the research proposed.

The Issue was addressed in lines 80-84

Background:

Lines 241, 246, 248 etc.: Once an abbreviation is introduced and defined, it should be consistently used thereafter instead of the full term.

Thank you for pointing this out. we addressed the multiple definitions of several terms such ML, GPG and BIM throughout the text.

Lines 103-304: While the section cites various studies, it tends to list them rather than critically analyze their contributions or limitations. A more in-depth analysis of these studies, including how they directly relate to the research proposed, would enhance the quality of the manuscript. Moreover, addressing how the proposed research builds upon or differs from these cited works would provide a clearer understanding of the paper's contribution to the field.

Thank you for the comment. This section was extensively edited revised, previous efforts for BIM element classification were critically reviewed and a more detailed description of the previous efforts was provided, where the knowledge gap was emphasized.  Please see section 2.1 in the manuscript. 

Lines 105-113, 185-192: Repetition should be avoided. For instance, the explanation of BIM and its challenges is revisited multiple times. Consolidating these explanations would streamline the content.

The repetition was addressed, some parts were discarded and the text reordered.

Methods:

Lines 437-453 : While the process of feature encoding and model training is described, it might be helpful to briefly discuss why certain methods (e.g., One Hot Encoding) were chosen over others.

The reasons for choosing this encoding were discussed in the methods section.

Lines 530-565: It would also be useful to clarify the reasoning behind the specific parameters and configurations used in the models (Logistic Regression, Random Forest, Serial Vector Machine, and K-Nearest Neighbor classifiers).

The reasons for selecting these models and their parameters are mainly related to their proved use in the literature. The relevant sources were added to the  text.

Figures: Lines 228, 317, 636, 645: Figure 2 is not referenced in the text. Additionally, the mention of a figure in the text should precede the figure itself. So please proceed to correct Figure 1. Last, there are two figures labeled as Figure 6 in the text.

Figure 2 is now mentioned in line 318.

Figure 1 was moved so that it appears after the mention.

We addressed the duplication of figure caption 6, however there seem to be some formatting issue in the document and we would request the magazine editors to make sure that the numbering is consistent

Results: Lines 611-621: The discussion on the variance in prediction accuracy across different test files is insightful but could benefit from deeper analysis. Why did these variances occur, and what implications does this have for the generalizability of the models?

We expanded the scope of this discussion in both sections 6.1 and 6.2. adding a diagram to better explain our observations. The implications regarding the generalizability of the model are indeed significant and we feel that our unique dataset is the first to surface this issue.

Discussion: Lines 664-715: While the section begins by summarizing the applicability of Graph Neural Networks, it would be beneficial to more directly connect these observations to the specific results presented earlier. For instance, discuss how the results support the advantages of using GNNs in BIM model classification. Also please elaborate on how the findings relate to existing literature.

The entire discussion section was restructured and revised to reflect these important comments.

Reviewer 3 Report

Comments and Suggestions for Authors

Dear Authors,

I appreciate the insightful work presented in your manuscript. To enhance the overall quality:

1.        Clarity and Conciseness: Streamline the manuscript for conciseness without sacrificing essential details. Ensure clarity in conveying your key points.

2.        English Expression: A thorough English review is recommended to refine the manuscript for fluency and coherence.

3.        Case Study Details: Provide more details on the case study or practical implications to enrich the reader's understanding of the real-world applications of your findings.

4.        Graph Structure: Elaborate on the specific graph structure used, its relevance, and how it captures contextual information. This would enhance the technical depth of your work.

5.        Comparative Analysis: In the results section, offer a more in-depth comparative analysis between classic ML models and Graph Neural Networks. Highlight specific scenarios where GNN outperforms classic ML models.

6.        Visual Aids: Consider including more visual aids, such as figures or tables, to illustrate key concepts or findings, making it more engaging for readers.

Best regards,

Comments on the Quality of English Language

Fair

Author Response

We thank the reviewer for these instructive comments. The text was edited accordingly, see answers to individual comments below.

  1. We edited the manuscript extensively, improving its structure. We tried to make the key points as clear and concise as possible.
  2. A through English review was conducted and many refinements and grammar corrections were made throughout the document. 
  3. We provide additional details regrading the case study, as well as  two more figures, one describing the. model itself and another about our findings. The conclusion section was expanded to include practical implications.
  4. An explanation about property graphs and their use in this research was added to section 4.1.2
  5. The comparison between the performance of the models was elaborated, both in the general case and in the case study. Figure 9, which describes the different behavior of the models in two specific scenarios was added.
  6.  As already mentioned, two additional figures (7 and 9) were added to the document to illustrate the case study and its conclusions.

Round 2

Reviewer 2 Report

Comments and Suggestions for Authors

The comments made in the previous review stage have been adequately addressed.

Reviewer 3 Report

Comments and Suggestions for Authors

Accept in present form

Comments on the Quality of English Language

Fair